# Different Features of Interleukin-37 and Interleukin-18 as Disease Activity Markers of Adult-Onset Still’s Disease

**DOI:** 10.3390/jcm10050910

**Published:** 2021-02-26

**Authors:** Seoung Wan Nam, SuMan Kang, Jun Hyeok Lee, Dae Hyun Yoo

**Affiliations:** 1Department of Rheumatology, Wonju Severance Christian Hospital, Yonsei University Wonju College of Medicine, Wonju 26426, Korea; namsw@yonsei.ac.kr; 2Hanyang University Institute for Rheumatology Research, Seoul 04763, Korea; fox1211@naver.com; 3Department of Biostatistics, Wonju College of Medicine, Yonsei University, Wonju 26426, Korea; ljh0101@yonsei.ac.kr; 4Department of Rheumatology, Hanyang University Hospital for Rheumatic Diseases, Seoul 04763, Korea

**Keywords:** interleukin-37, interleukin-18, adult-onset Still’s disease

## Abstract

The aim of this study was to evaluate the usefulness of serum interleukin (IL)-37 and IL-18 as disease activity markers of adult-onset Still’s disease (AOSD) and to compare their related clinical features. Forty-five patients with a set of high and subsequent low disease activity status of AOSD were enrolled. Modified Pouchot (mPouchot) score and serologic disease activity markers including levels of IL-37 and IL-18 were compared between high and low disease activity status. The relationships between disease activity parameters and differences in levels of cytokines according to each disease manifestation were evaluated in high disease activity status. mPouchot score and all disease activity markers including IL-37 and IL-18 significantly declined after treatment. Though both cytokines positively correlated with mPouchot score, the two did not correlate with each other in high disease activity status. IL-18 positively correlated with ferritin, AST, and LDH while IL-37 correlated better with CRP. The expression level of IL-37 was related to leukocytosis while IL-18 was related to pleuritis, pneumonitis, abnormal LFT, and hyperferritinemia. In addition, patients in the IL-18 dominant group presented with higher LDH levels and required a higher mean corticosteroid dose. In conclusion, IL-37 and IL-18 are disease activity markers reflecting different aspects of AOSD that can complement each other.

## 1. Introduction

Adult-onset Still’s disease (AOSD) is a rare systemic auto-inflammatory disorder characterized by four cardinal symptoms of spiking fever, arthralgia or arthritis, evanescent salmon-colored rash, and leukocytosis with neutrophil predominance [1]. The diagnosis of AOSD is still challenging owing to its heterogeneous clinical manifestations and few disease-specific markers [2]. Likewise, it is difficult to assess disease activity of AOSD. Pouchot et al. described a systemic disease activity score comprised of 12 main signs and symptoms of AOSD; this scoring system was later modified by Rau et al. [3,4]. Several biomarkers were proposed as potential disease activity parameters of AOSD including C-reactive protein (CRP), ferritin, and interleukin (IL)-18 [1,2,5]. However, no reliable disease activity parameter exists yet [6]. Given the potential polycyclic evolution of AOSD, it is necessary to identify biomarkers for accurate assessment of disease activity to improve AOSD management [1].

Although the pathogenesis of AOSD is unclear, many proinflammatory cytokines, such as IL-1β, IL-6, tumor necrosis factor (TNF)-α, interferon (IFN)-γ, and IL-18 seem to be involved; levels of these cytokines positively correlated with disease activity [6]. Among these cytokines, IL-18 is the most promising biomarker of AOSD as its serum level is particularly high in AOSD compared to other inflammatory diseases such as rheumatoid arthritis (RA), ankylosing spondylitis (AS), polymyalgia rheumatica, and sepsis [1,2,7,8]. IL-18 has not only been suggested as a potential biomarker for differential diagnosis of AOSD, but also for evaluation of disease activity [1,2]. However, its use in routine investigations has not been widely accepted as evidence of its reliability is insufficient [1,6,9].

IL-37 is a newly discovered member of the IL-1 cytokine family and has been identified as an inhibitor of immune responses in contrast to IL-18, another member of the IL-1 family [10]. Studies have shown elevated expression of IL-37 and its positive correlations with the disease activity parameters in patients with autoimmune diseases such as RA, systemic lupus erythematosus, AS, and systemic juvenile idiopathic arthritis [11,12,13,14,15]. In addition, a recent study identified increased expression of IL-37 and its positive correlation with disease activity markers in patients with AOSD [16]. However, our knowledge about the role of IL-37 in the pathogenesis of AOSD and its clinical applicability as a biomarker of the disease is very limited. Moreover, no study has compared IL-37 and IL-18 as disease activity parameters in patients with AOSD.

In this study, we evaluated how serum levels of IL-37 and IL-18 reflect the activity of AOSD using paired serum samples of high and low disease activity status in a cohort of patients with AOSD.

## 2. Materials and Methods

### 2.1. Study Population and the Evaluation Period

Sixty patients with AOSD were enrolled in this study, all of whom met the criteria of Yamaguchi et al. Four major criteria proposed by Yamaguchi include fever, arthralgia or arthritis, non-pruritic salmon-colored rash, and leukocytosis with granulocyte predominance. Minor criteria include sore throat, lymphadenopathy, hepatomegaly and/or splenomegaly, abnormal liver function, and negative test results for antinuclear antibody and rheumatoid factor. At least five criteria, including two major criteria and no exclusion criteria (infection, malignancy, and other rheumatic diseases), are required for the diagnosis of AOSD (Appendix A) [17]. Those with concurrent infection, malignancies, or other rheumatic diseases were not eligible. The medical records of patients were reviewed thoroughly regarding AOSD activity status assessed by the modified Pouchot (mPouchot) score proposed by Rau et al. [3,4]. The total score that ranges from 0 to 12 is calculated by assigning one point to each of these items: the presence of fever, evanescent rash, sore throat, arthritis, myalgia, pleuritis, pericarditis, pneumonitis, lymphadenopathy, hepatomegaly or abnormal liver function tests, white blood cell count (WBC) > 15,000/mm^3^, and serum ferritin > 3000 ng/mL. Consistent with previous studies, a high disease activity status of AOSD was defined as mPouchot score ≥ 4 [4,8]. Low disease activity status of AOSD was defined arbitrarily as mPouchot score ≤ 2 [18]. A set of high and subsequent low disease activity statuses that fall within a 12-month period were determined in each patient. For comparison of clinical parameters between high and low disease activity status of AOSD, patients without a record of definite high disease activity status or those with a high and low disease activity status interval longer than 12 months were excluded from the study.

A retrospective analysis of AOSD patients who prospectively visited a single university hospital for periodic examination or admission between August 2016 and April 2017 was performed. This analysis was an evaluation of clinical information from the initial diagnosis of AOSD until the time of investigation, April 2017. The study protocol was approved by the Institutional Review Board of Hanyang University Hospital, and informed consent was received from all participants (IRB no. HYUH 2016-06-007).

### 2.2. Laboratory Studies

Serial serum samples were collected from all study patients at one- to three-month intervals depending on the clinical situation. In addition to routine clinical laboratory studies including complete blood cell count and liver function tests, disease activity parameters such as ferritin, interleukin (IL)-18, and IL-37 levels were measured. Serum samples were obtained in the morning after overnight fasting, and care was taken to avoid hemolysis. The serum ferritin level was analyzed using the electrochemiluminescence method (Cobas e 601, Roche Diagnostics, Mannheim, Germany). Erythrocyte sedimentation rate (ESR) level was measured using the Westergren method (Starrsed, R & R mechatronic, Zwaag, The Netherlands). The AU5822 automated clinical chemistry analyzer (Beckman Coulter, Brea, CA, USA) was used for the measurement of levels of C-reactive protein (CRP), aspartate aminotransferase (AST) and alanine aminotransferase (ALT), lactate dehydrogenase (LDH) using the immunoturbidimetric method (CRP), UV method (AST and ALT), and lactate to pyruvate method (LDH).

For measurement of cytokines, samples were centrifuged at 2500 rpm for 15 min at 5 °C within 30 min of collection at room temperature and maintained at −70 °C until use. Serum IL-18 (Medical and Biological Laboratories, Nagoya, Japan) and IL-37 (Elabscience Biotechnology Inc., Huston, TX, USA) were measured using commercial enzyme-linked immunosorbent assay kits, following the manufacturers’ instructions. If the absorbance of a sample for the IL-18 measurement exceeded 2500 pg/mL, the sample was diluted 20–200-fold. And, if the sample for the IL-37 measurement exceeded 1000 pg/mL, it was diluted 10-fold. The researcher who read the ELISA measurements was blinded to the patient information [2].

### 2.3. Statistical Analysis

Data are expressed according to the properties of the variables. The mPouchot score, clinical parameters included in the mPouchot score, and laboratory parameters were compared between high and low disease activity status using the McNemar test, Exact McNemar test, or Wilcoxon signed-rank test, as appropriate.

Clinical characteristics and laboratory parameters were analyzed in high disease activity statuses of AOSD. The relationships among the disease activity parameters were evaluated using Spearman’s correlation coefficients. The patients were divided into two groups, the IL-37 dominant group (patients with IL-37 ≥ median and IL-18 < median) and the IL-18 dominant group (patients with IL-18 ≥ median and IL-37 < median); clinical characteristics and laboratory parameters were compared using Wilcoxon rank-sum test and Fisher’s exact test, as appropriate. Serum levels of IL-37 and IL-18 were compared according to the presence or absence of each disease manifestation of AOSD by the Wilcoxon rank-sum test.

All statistical analyses were performed using SAS software version 9.4 (SAS, Cary, NC, USA) and R version 3.5.3. (The R Foundation for Statistical Computing, Vienna, Austria).

## 3. Results

Among the 60 patients recruited for this study, 45 were included. Nine patients did not show definite high disease activity defined as mPouchot score ≥ 4, and six patients did not cycle between high and low disease activity status within a 12-month period; these 15 were excluded from the study.

### 3.1. Demographic and Clinical Characteristics

The demographic and clinical characteristics of patients with AOSD included in this study are shown in Table 1. The overall mean age of the patients was 47.1 (±14.0) years, with a mean disease duration of 59.5 (±48.2) months; female patients were more prevalent (86.7%). Retrospective analyses of the medical records of all patients revealed that the most common AOSD onset symptom was a combined pattern of systemic and articular diseases (68.9%). A total of 57.8% of patients showed two or more systemic disease flare-ups by April 2017, the end of the observation period.

The mean interval between high disease activity status and subsequent low disease activity status was 6.0 (±2.7) months. During this period, 17.8% of patients underwent steroid pulse therapy, and the most frequently used immune-modulating medications were methotrexate (77.8%), cyclosporin A (40.0%), and leflunomide (22.2%). Approximately one-third of patients (33.3%) had a re-flare of AOSD within one year after a low disease activity status, with a mean duration until re-flare of 4.3 (±4.8) months. More than one-half of patients (54.3%) had a re-flare within two years after achieving low disease activity status.

### 3.2. Paired Comparison of Clinical Features and Laboratory Parameters between High and Low Disease Activity Status

As shown in Table 2, the mPouchot score and all the laboratory parameters investigated, including IL-37 and IL-18, were significantly lower in the subsequent low disease activity status compared to the high disease activity status. The most common disease manifestations of AOSD in high disease activity status were fever (86.7%), skin rash (82.2%), arthritis (73.3%), myalgia (68.9%), and sore throat (66.7%). The least frequent clinical findings were pleuritis (13.3%) and pericarditis (4.4%), which failed to show a significant difference between high and low disease activity states.

### 3.3. Correlations between Disease Activity Parameters in High Disease Activity Status

As shown in Table 3, IL-37 and IL-18 showed different patterns of correlations with other serologic disease activity markers in high disease activity status of AOSD. Unlike IL-18, which correlated better with serum levels of ferritin, LDH, and AST, IL-37 correlated better with CRP. IL-37 and IL-18 showed comparable correlations with systemic disease activity score (mPouchot score) (Spearman’s rho = 0.355 and 0.399, respectively, *p* < 0.05). However, no association was established between IL-37 and IL-18.

### 3.4. IL-37 and IL-18 in High Disease Activity Status of AOSD

As shown in Figure 1, serum IL-37 and IL-18 levels were diffusely scattered at high disease activity status of AOSD, showing no linear correlation. Clinical characteristics were compared between two groups, each with 10 patients, with contrasting serum concentration levels of IL-37 and IL-18 (Table 4, Appendix A). There were no significant differences between the IL-37 dominant group and the IL-18 dominant group in demographics, AOSD disease course, incidence of disease flares, use of immune-modulating drugs at high disease activity status, and re-flare rate after low disease activity status. However, the IL-18 dominant group showed higher LDH levels at high disease activity status, and these patients required a higher mean steroid dose for achieving low disease activity status (*p* = 0.01 and 0.02, respectively). However, there was no difference in mPouchot score between the two groups in high disease activity status.

As shown in Table 5, IL-37 and IL-18 serum levels showed different patterns of increase according to the presence of each disease manifestation of AOSD at high disease activity status. Serum IL-18 levels were higher in patients with pleuritis, pneumonitis, abnormal liver function test, and hyperferritinemia (*p* < 0.05). In contrast, serum IL-37 level was only higher in patients with leukocytosis (*p* < 0.01). There was no concurrent increase in serum levels of IL-37 and IL-18 related to the presence of any clinical domain of AOSD.

## 4. Discussion

AOSD is a systemic auto-inflammatory disorder that presents with heterogeneous clinical manifestations. Its pathogenesis is largely unknown, and no single efficient disease activity marker for this disease has been identified [1,2]. Regarding its heterogeneity in the clinical presentation, we might need multi-modal approaches for investigating pathogenic mechanisms and assessing disease activity. In this study, we demonstrated that IL-37 level reflects disease activity of AOSD, as does the known cytokine biomarker IL-18, but in different ways [1].

IL-37 and IL-18 comprise the IL-18 subfamily of the IL-1 family as both bind to the IL-18 receptor [19,20]. IL-18 plays a key role in the polarized type 1 innate and adaptive responses that can possibly extend to the mediation of autoimmune diseases [21]. In contrast, IL-37 is known to be negatively involved in the development and pathogenesis of autoimmune diseases [16,22,23]. The expression level of IL-37 is low in healthy human tissues but is stimulated in severe inflammatory conditions to inhibit excessive immune response [22,24]. Proinflammatory stimuli including cytokines such as IL-1β or engagement of various Toll-like receptors activate the production of IL-37 [10,25]. Though both IL-18 and IL-37 bind to the α chain of the IL-18 receptor (IL-18Rα) on the cell surface, the IL-37 combined complex transduces anti-inflammatory signals while the IL-18 combined complex initiates a strong proinflammatory process that produces IFN-γ [10,26,27]. IL-18 induces the production of IFN-γ, IL-17A, and TNF-α, which play an important role in the disease manifestations of AOSD [28]. IL-6 is also elevated and known to be related to some clinical features of AOSD such as fever, arthritis, and increased production of acute-phase proteins [29,30]. Circulating IL-18 Binding Protein (IL-18BP) provides the main regulatory mechanism on IL-18 mediated inflammation by sequestrating IL-18 and prevents its binding to the receptor [10]. IL-18BP also binds to IL-37 and hence high level of IL-18BP could reduce the anti-inflammatory activity of IL-37. In addition, IL-37 can translocate into the nucleus and regulate gene expression of proinflammatory cytokines such as TNF-α, IL-1α, and IL-6 [10]. Previous studies conducted in murine models revealed the improvement of clinical features related to inflammation such as fatigue, arthritis, insulin resistance, and gouty arthritis after treatment with recombinant IL-37 [31,32,33,34]. Gout is another auto-inflammatory disease that shares the common pathophysiology of IL-1 mediated inflammation with AOSD. Joosten, et al. previously identified rare genetic variants of IL-37 in gout patients by sequencing all coding bases of IL-37 [34]. These results all indicate the potential mechanistic role of IL-37 in the pathogenesis of AOSD.

Previous studies supported the usefulness of IL-18 in AOSD differential diagnosis, disease activity evaluation, subset prediction (systemic or chronic articular subtypes), and severity assessment [1,2,5,7,8,18]. However, there is scarce information about the role of IL-37 in AOSD [16]. Chi et al. recently reported that the serum level of IL-37 was higher in patients with AOSD compared to healthy subjects, and a significant difference in the serum level of IL-37 was observed between patients with active and inactive disease status [16]. The serum level of IL-37 was even higher in patients with inactive disease compared to healthy subjects. In addition, their results showed a significant decrease in serum level of IL-37 after treatment in 10 patients with serial work-up data [16]. Though previous studies suggested both IL-37 and IL-18 as estimators of disease activity in patients with AOSD, controversies exist as to whether IL-18 could serve as a marker for disease remission and follow-up [1]. We noted four studies that demonstrated changes in serum level of IL-18 between high and subsequent low disease activity status after treatment [2,18,35,36]. Two studies from the same institution showed no significant decrease in serum IL-18 level after treatment in 16 and 17 patients with AOSD, respectively [18,35]. However, they did not apply specific criteria for a high disease activity status of AOSD. Contrary to their results, our previous study indicated that serum IL-18 declined significantly in 18 patients with AOSD who responded to therapy but not in non-responders [2]. In addition, the serum IL-18 level was higher in patients with low disease activity compared to those in remission, unlike other serologic disease activity markers. Therefore, we previously suggested IL-18 as an efficient marker for remission and follow-up in patients with AOSD [2]. In the current study, the serum IL-18 level decreased significantly in patients who achieved low disease activity status along with other disease activity markers including leukocytes, ESR, CRP, AST, ALT, ferritin, and IL-37 (Table 2). Our results are also compatible with those of another report of 11 patients with AOSD [36]. The differing results regarding change of serum IL-18 level could be due to different criteria for high and low disease activity status or differences in baseline characteristics of study populations, such as disease subset type. Previous studies have demonstrated that serum IL-18 level tends to be higher in patients presenting with systemic subset compared to chronic articular subtype [1,37]. After applying clear set criteria for both high and low disease activity status, we noted that both IL-37 and IL-18 could be used as disease activity evaluators even for follow-up of AOSD activity.

We evaluated IL-37 and IL-18 in relation to other disease activity parameters and various clinical features in high disease activity status of AOSD to compare their roles in the activity of this disease. In addition to there being no correlation between IL-18 and IL-37 serum levels, the differences in their patterns of positive correlations with other serologic markers suggest their different pathogenic roles in disease activity (Table 3). In addition, there were different patterns in the increase of both serum cytokines regarding the presence of each clinical manifestation of AOSD (Table 5). IL-37 seems to be more related to non-specific markers of inflammation such as CRP and leukocytes [1]. In contrast, IL-18 seems to be more related to pleuritis, pneumonitis, and serologic markers associated with liver injury such as AST, ALT, ferritin, and LDH [7,30,38,39,40]. We also demonstrated that some patients with AOSD could be grouped as those with the dominant increase in the serum level of IL-37 or IL-18 (Figure 1). Though patients in both groups showed no difference in mPouchot score, those in the IL-18 dominant group presented with higher serum level of LDH at high disease activity status. In addition, they required a higher mean corticosteroid dose to achieve low disease activity status with no difference in the cumulative dosage of corticosteroid (Table 4 and Appendix A). LDH performs a prominent role in the active metabolism of the overall body and unusual LDH isoform level in serum serves as a significant biomarker of different diseases [41]. In a case report, Motoo et al. demonstrated that the increased serum LDH during an AOSD flare-up was mainly of liver origin (LDH isoenzymes 4 and 5) [40]. Macrophage-derived IL-18 plays a fundamental role in hepatocyte apoptosis via activation of the Fas/Fas ligand pathway [39,42,43]. Priori et al. demonstrated that the serum IL-18 level is markedly increased in patients with AOSD-related hepatitis. In addition, they histologically revealed that IL-18 was highly expressed by activated CD68+ liver macrophages and Kupffer cells in a patient with AOSD [38]. Therefore, it is possible that IL-18 related hypermetabolic changes or hepatic injury could have led to a higher intensity of therapy in IL-18 dominant group patients in the current study. It has been known that hyperferritinemia in AOSD patients is primarily due to increased secretion in the liver and spleen in patients with AOSD [7,30]. CRP is produced only by hepatocytes, predominantly induced by IL-6, in response to inflammation [44]. However, its increase in serum could be dampened by severe liver dysfunction [45]. In contrast, the increase in serum level of IL-18 or ferritin could reflect the status of liver injury [1,46,47,48]. Patients with AOSD present with heterogeneous clinical features of multi-systemic involvement [29]. Likewise, not all patients in this study presented with an abnormal liver function test or serum ferritin level higher than 3000 ng/mL in high disease activity status (Table 2). Therefore, other serologic markers could be better markers of disease activity compared to IL-18 or ferritin in a substantial portion of patients with AOSD. For a better evaluation of disease activity in patients with AOSD, we might need additional useful biomarkers or a multi-biomarker disease activity index that properly reflects a patient status such as what has developed for patients with RA [49].

In all relevant previous studies regarding correlations between serum level of IL-18 and other serologic markers in patients with active AOSD, serum IL-18 level positively correlated with serum ferritin level and systemic disease scores of AOSD, alike our study results [7,8,18,35,38,50]. However, its correlations with serum ESR or CRP have shown inconsistent results, probably due to the different characteristics of patients studied [7,8,18,35,38]. Only two studies conducted by Priori et al. showed significant positive correlations between serum IL-18 level and non-specific inflammatory markers. However, unlike ours, their analyses included patients in inactive disease status of AOSD [8,38]. According to a previous study of IL-37 in patients with AOSD conducted by Chi et al., serum IL-37 level positively correlated with non-specific disease activity markers including leukocytes and CRP but not with liver enzymes. However, unlike our study results, serum IL-37 level correlated well with IL-18 level in 62 patients with AOSD [16]. In addition, more diverse clinical features (fever, sore throat, rash, lymphadenopathy, splenomegaly, myalgia, and arthralgia) were significantly related to higher levels of IL-37. We noted that a third of their study population (21 patients) were inactive AOSD patients with a mean mPouchot score of 0.2 ± 0.5 indicating almost no disease-related symptoms. Therefore, the difference between the two groups could have been more easily reflected in their analysis results owing to the conflicting disease activities among patients [51]. With very limited information especially regarding the role of IL-37 in AOSD, we are unaware whether how differently both IL-37 and IL-18 change in their serum concentrations depending on the disease activity of AOSD [22]. Therefore, to examine the correlation between both cytokines in patients with AOSD, it would be better to control the disease activity of AOSD in the analysis to get more reliable results.

The current study and previous relevant study findings, overall, indicate that both serum levels of IL-37 and IL-18 can be good markers of disease activity in patients with AOSD. We add new finding by demonstrating different patterns of increase in serum levels of both cytokines and related clinical features in the high disease activity status of AOSD. Measuring both serum levels of IL-37 and IL-18 could complement each other to better understand this heterogeneous, multi-systemic disease. Our study has several limitations. First, the sample size of this study was small owing to the rarity of the disease. However, our study was the largest comparing paired serum samples of both high and low disease activity statuses of AOSD. Second, our evaluation did not include other potential mechanistic cytokines of AOSD, such as IL-1β, IL-6, or TNF-α [1]. Future studies with larger sample sizes that evaluate various serologic cytokine profiles are needed. Third, there could be the concern that the use of immune-modulating drugs at high disease activity status could have affected patient cytokine profile to some extent. However, we noted that there were no significant differences in the frequencies of immune-modulating drug use between compared groups of Table 4 and Table 5. Fourth, as our study was a retrospective observational study, patient visit interval or therapeutic protocol was not constant among patients. Study patients were treated by a single rheumatologist for the purpose of controlling disease activity with optimal medical therapies available. The patient visit interval was determined by clinical situation and varied from two weeks to three months during the observation period. Last, there was no consensus in defining high or low disease activity status of AOSD or in defining re-flare of disease after achieving low disease activity status. Therefore, we referred to previous studies that applied similar definitions [4,8,18]. However, we applied the operational definition of disease re-flare as an increase in mPouchot score, increase in dosage, or addition of immune-modulating drugs including glucocorticoids.

## 5. Conclusions

We identified that serum levels of IL-37 and IL-18 reflect different clinical features of AOSD. Both could be utilized as disease activity markers that can complement each other.

## Figures and Tables

**Figure 1 jcm-10-00910-f001:**
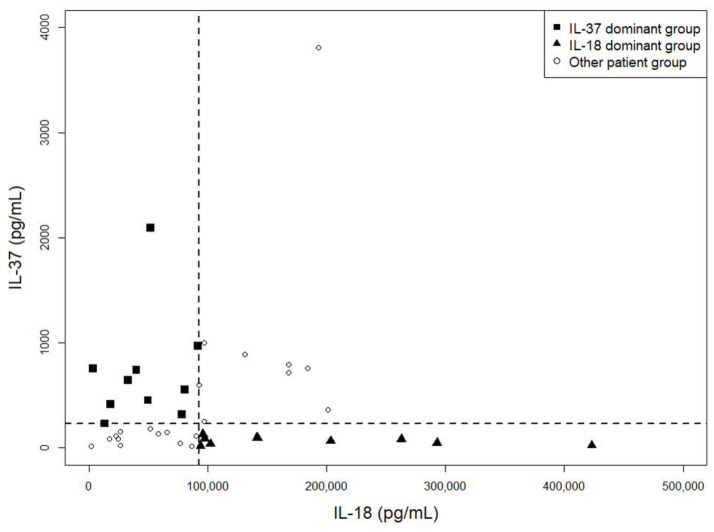
Scatter plot of IL-37 and IL-18 levels at a high disease activity status of adult-onset Still’s disease. Serum IL-37 and IL-18 levels were diffusely scattered at a high disease activity status of AOSD, showing no linear correlation. Each symbol represents an individual patient and dotted lines represent the median values of IL-37 and IL-18. AOSD, adult-onset Still’s disease; IL, interleukin.

**Table 1 jcm-10-00910-t001:** Demographic and clinical characteristics of patients with adult-onset Still’s disease ^a^.

	Total Patients (*n* = 45)
Age	47.1 ± 14.0
Female, n (%)	39 (86.7)
Disease duration (months)	59.5 ± 48.2
Number of Yamaguchi criteria met at diagnosis	6.3 ± 1.0
AOSD onset symptom, n (%)	
Systemic	11 (24.4)
Articular	3 (6.7)
Combined	31 (68.9)
Number of disease flares (per year)	0.6 ± 0.4
Disease duration until high disease activity status (months)	28.9 ± 43.3
Interval between high and low disease activity status (months)	6.0 ± 2.7
Medications used during the evaluation period, n (%)	
Corticosteroids	45 (100)
Cumulative corticosteroid dose (mg) ^b^	2769.7 ± 1511.0
Mean corticosteroid dose (mg/day) ^b^	17.1 ± 9.5
Corticosteroid pulse therapy ^c^	8 (17.8)
Methotrexate	35 (77.8)
Leflunomide	10 (22.2)
Cyclosporin A	18 (40.0)
Azathioprine	2 (4.4)
Hydroxychloroquine	5 (11.1)
IV immunoglobulin	1 (2.2)
Tocilizumab	3 (6.7)
Patients with re-flare after low disease activity status	
Re-flare within 1 year (missing = 3), n (%)	14 (33.3)
Duration until re-flare (months)	4.3 ± 4.8
Re-flare within 2 years (missing = 10), n (%)	19 (54.3)
Duration until re-flare (months)	8.3 ± 7.5
Death, n (%)	1 (2.2)

AOSD, adult-onset Still’s disease; IV, intravenous. ^a^ Values are shown as mean ± standard deviation unless otherwise indicated. ^b^ Values represent mg of prednisone equivalent and do not include the dose of corticosteroid pulse therapy. ^c^ Corticosteroid pulse therapy is defined as the administration of ≥250 mg prednisone equivalent per day for one or a few days.

**Table 2 jcm-10-00910-t002:** Paired comparison of clinical characteristics between high and low disease activity status in patients with adult-onset Still’s disease ^a^.

	High Disease Activity (mPouchot Score ≥ 4)	Low Disease Activity (mPouchot Score ≤ 2)	*p*
Modified Pouchot score ^b^	6.0 (4.5–7.0)	0 (0–0)	<0.01
Fever	39 (86.7)	0 (0)	<0.01
Evanescent rash	37 (82.2)	2 (4.4)	<0.01
Sore throat	30 (66.7)	0 (0)	<0.01
Arthritis	33 (73.3)	5 (11.1)	<0.01
Myalgia	31 (68.9)	0 (0)	<0.01
Pleuritis	6 (13.3)	0 (0)	0.03
Pericarditis	2 (4.4)	1 (2.2)	0.32
Pneumonitis	12 (26.7)	0 (0)	<0.01
Lymphadenopathy	14 (31.1)	0 (0)	<0.01
Hepatomegaly or abnormal LFT ^c^			
Hepatomegaly	4 (8.9)	0 (0)	0.13
Abnormal LFT ^c^	29 (64.4)	3 (6.7)	<0.01
WBC > 15,000 (10^9^/L)	19 (42.2)	2 (4.4)	<0.01
Ferritin > 3000 (ng/mL)	22 (48.9)	0 (0)	<0.01
Laboratory findings ^b^			
WBC (10^9^/L)	12.8 (7.4–20.0)	8.0 (6.9–10.1)	<0.01
ESR (mm/hr)	79.5 (34.0–101.0)	13.0 (8.0–32.0)	<0.01
CRP (mg/dL)	7.5 (3.2–12.7)	0.4 (0.2–0.4)	<0.01
AST (U/L)	38.0 (26.0–56.0)	20.0 (16.0–23.0)	<0.01
ALT (U/L)	32.0 (17.0–56.0)	16.0 (12.0–19.5)	<0.01
LDH (U/L)	368.0 (285.0–489.0)	198.0 (161.0–240.0)	<0.01
Ferritin (ng/mL)	2628.0 (538.4–8791.0)	59.3 (35.4–108.1)	<0.01
IL-18 (pg/mL)	92,874.2 (49,764.2–141,944.5)	1316.0 (666.8–6069.9)	<0.01
IL-37 (pg/mL)	231.6 (86.5–754.0)	29.2 (7.5–64.0)	<0.01

mPouchot, modified Pouchot; LFT, liver function test; WBC, white blood cells; ESR, erythrocyte sedimentation rate; CRP, C-reactive protein; AST, aspartate aminotransferase; ALT, alanine aminotransferase; LDH, lactate dehydrogenase; IL, interleukin. ^a^ Values are numbers (percentage) unless otherwise indicated. ^b^ Median (Q1–Q3) ^c^ Defined as an abnormal value of AST, ALT, or alkaline phosphatase.

**Table 3 jcm-10-00910-t003:** Correlations between disease activity parameters in high disease activity status of adult-onset Still’s disease.

	mPouchot Score	ESR	CRP	Ferritin	AST	ALT	LDH	IL-18	IL-37
mPouchot score	1	0.515 **	0.481 **	0.616 **	0.247	0.228	0.236	0.399 **	0.355 *
ESR	0.515 **	1	0.418 **	0.218	−0.140	−0.04	−0.340 *	−0.126	0.270
CRP	0.481 **	0.418 **	1	0.424 **	0.09	−0.09	0.079	0.287	0.573 **
ferritin	0.616 **	0.218	0.424 **	1	0.543 **	0.343 *	0.564 **	0.656 **	0.336 *
AST	0.247	−0.140	0.090	0.543 *	1	0.559 **	0.640 **	0.468 *	0.026
ALT	0.228	−0.040	−0.087	0.343 *	0.559 **	1	0.263	0.238	−0.088
LDH	0.236	−0.340*	0.079	0.564 **	0.640 **	0.263	1	0.741 **	0.153
IL-18	0.399 **	−0.126	0.287	0.656 **	0.468 **	0.238	0.741 **	1	0.159
IL-37	0.355 *	0.270	0.573 **	0.336 *	0.03	−0.088	0.153	0.159	1

By Spearman’s correlation coefficients. * Correlation is significant at the 0.05 level. ** Correlation is significant at the 0.01 level. mPouchot, modified Pouchot; ESR, erythrocyte sedimentation rate; CRP, C-reactive protein; AST, aspartate aminotransferase; ALT, alanine aminotransferase; LDH, lactate dehydrogenase; IL, interleukin.

**Table 4 jcm-10-00910-t004:** Comparison of disease activity parameters and medications used between the IL-37 and IL-18 dominant groups in patients with adult-onset Still’s disease ^a^.

	IL-37 Dominant Group (*n* = 10)	IL-18 Dominant Group (*n* = 10)	*p*
Disease activity parameters at high disease activity status			
Modified Pouchot score	7 (6–7)	6 (6–7)	0.97
Laboratory findings			
WBC (10^9^/L)	16.4 (8.4–24.0)	10.8 (6.3–17.0)	0.25
ESR (mm/hr)	102.5 (62.0–115.0)	73.0 (36.0–83.0)	0.09
CRP (mg/dL)	8.8 (5.3–15.0)	4.1 (2.8–7.6)	0.11
AST (U/L)	30.5 (22.0–40.0)	42.0 (29.0–69.0)	0.17
ALT (U/L)	23.5 (12.0–49.0)	45.0 (17.0–90.0)	0.12
LDH (U/L)	302.0 (246.0–354.0)	492.0 (355.0–522.0)	0.01
Ferritin (ng/mL)	2069.5 (727.0–2628.0)	5246.5 (2268.0–11254.0)	0.11
Medications used during the evaluation period			
Corticosteroids, n (%)	10 (100)	10 (100)	1.00
Cumulative corticosteroid dose (mg) ^b^	2590.0 (1675.0–3155.0)	2672.5 (2000.0–3307.5)	0.80
Mean corticosteroid dose (mg/day) ^b^	12.9 (10.7–23.7)	24.6 (17.3–26.4)	0.02
Corticosteroid pulse therapy^c^, n (%)	4 (40.0)	0 (0.0)	0.09
Methotrexate, n (%)	8 (80.0)	7 (70.0)	1.00
Leflunomide, n (%)	1 (10.0)	5 (50.0)	0.14
Cyclosporin A, n (%)	3 (30.0)	5 (50.0)	0.65

IL, interleukin; WBC, white blood cells; ESR, erythrocyte sedimentation rate; CRP, C-reactive protein; AST, aspartate aminotransferase; ALT, alanine aminotransferase; LDH, lactate dehydrogenase. ^a^ Values are median (Q1-Q3) unless otherwise indicated. ^b^ Values represent mg of prednisone equivalent and do not include the dose of corticosteroid pulse therapy. ^c^ Corticosteroid pulse therapy is defined as the administration of ≥250 mg prednisone equivalent per day for one or a few days.

**Table 5 jcm-10-00910-t005:** Comparison of serum IL-37 and IL-18 levels according to disease manifestations at high disease activity status of patients with adult-onset Still’s disease.

Disease Manifestation	*n*	Serum IL-37 (pg/mL) ^a^	*p*	Serum IL-18 (pg/mL) ^b^	*p*
Fever	(+), n = 39	320.6 (87.1–794.9)	0.09	95,879.0 (51,460.0–168,103.9)	0.14
	(−), n = 6	88.3 (24.4–150.0)		46,295.9 (22,779.3–86,407.4)	
Skin rash	(+), n = 37	231.6 (94.5–740.1)	0.58	90,140.0 (39,970.0–131,420.0)	0.14
	(−), n = 8	339.3 (30.2–857.2)		143,178.8 (72,357.1–342,925.0)	
Sore throat	(+), n = 30	434.4 (97.1–794.9)	0.10	88,979.8 (32,665.4–131,420.0)	0.14
	(−), n = 15	114.4 (47.7–365.2)		95,879.0 (76,861.7–203,430.2)	
Arthritis	(+), n = 33	185.0 (86.5–740.1)	0.34	90,140.0 (39,970.0–131,420.0)	0.15
	(−), n = 12	409.4 (99.0–2383.6)		116,849.2 (57,839.5–232,115.6)	
Myalgia	(+), n = 31	157.4 (82.5–740.1)	0.53	80,626.0 (32,665.4–141,081.1)	0.40
	(−), n = 14	409.4 (86.5–794.9)		94,815.6 (86,407.3–184,044.8)	
Pleuritis	(+), n = 6	444.7 (64.3–5160.0)	0.73	154,595.5 (131,385.5–203,430.2)	0.02
	(−), n = 39	231.6 (86.5–740.1)		86,407.3 (32,665.4–131,420.0)	
Pericarditis	(+), n = 2	532.1 (64.3–1000.0)	0.89	150,216.7 (97,003.2–203,430.2)	0.26
	(−), n = 43	231.6 (86.5–754.0)		91,552.2 (39,970.0–141,944.5)	
Pneumonitis	(+), n = 12	504.2 (172.6–978.6)	0.17	116,849.2 (88,691.5–184,742.5)	0.05
	(−), n = 33	150.0 (85.5–716.0)		86,407.3 (26,677.0–131,420.0)	
Lymphadenopathy	(+), n = 14	409.4 (157.4–794.9)	0.14	131,402.8 (78,019.7–168,103.9)	0.12
	(−), n = 31	137.7 (64.3–740.1)		86,407.3 (32,665.4–102,312.8)	
Hepatomegaly	(+), n = 4	458.4 (170.8–776.6)	0.89	93,422.0 (85,447.0–874,084.9)	0.49
	(−), n = 41	185.0 (86.5–754.0)		91,552.2 (39,970.0–141,944.5)	
Abnormal LFT	(+), n = 29	231.6 (85.5–740.1)	0.86	97,003.2 (65,914.8–168,103.9)	0.03
	(−), n = 16	239.0 (90.5–755.6)		67,616.7 (22,186.5–97,900.1)	
WBC > 15,000 (10^9^/L) ^c^	(+), n = 19	716.0 (231.6–972.9)	<0.01	97,003.2 (51,840.0–168,103.9)	0.36
	(−), n = 26	123.5 (64.3–320.6)		82,213.5 (26,677.0–141,081.1)	
Ferritin > 3000 (ng/mL) ^d^	(+), n = 22	656.1 (82.5–957.1)	0.18	154,592.5 (96,757.0–203,430.2)	<0.01
	(−), n = 23	150.0 (87.1–453.5)		58,371.8 (22,779.3–90,140.0)	

IL, interleukin; LFT, liver function test; WBC, white blood cells. Values are median (Q1-Q3). ^a^ Reference value of serum IL-37 was unavailable. ^b^ Reference range was 36.1–257.8 pg/mL according to the manufacturer’s instructions of enzyme-linked immunosorbent assay kit. ^c^ Reference: 4000–10,000 (10^9^/L). ^d^ Reference: 10–291 ng/mL.

## Data Availability

The data presented in this study are available on request from the corresponding author. The data are not publicly available due to the ethical issues.

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
