# Peer review of "Different Features of Interleukin-37 and Interleukin-18 as Disease Activity Markers of Adult-Onset Still’s Disease"

_jcm, 2021, doi:10.3390/jcm10050910_

Round 1

Reviewer 1 Report

Introduction:

Lines 55-59: It seems redundant; modify the sentence for clarity.

 Results:

Fig. 1. What do the empty circles represent?

Table 4: What are the ranges for healthy controls?

Discussion:

Lines 221-224: Reference for Chi et al. is missing.

Discussion on the roles of common inflammatory cytokines such as IL-1, IL-4, IL-13, IL-17A, IFN-gamma, and TNF-alpha in AOSD is missing.

Reviewer 2 Report

This is an interesting clinical report, which at present lack some pathophysiologic insight.  The discussion reports redundant information that is already presented in the introduction. I suggest that the authors discuss some pathophysiology of IL-37, which would help interpreting the correlations with disease activity. For example, were the Authors aware that IL-37 binds to the alpha chain of the IL-18 receptor, and that it also binds IL-18 binding protein (IL-18BP), the main regulatory mechanism preventing IL-18-mediated inflammation? This is important cytokine biology which must be discussed in this paper (see for example the review: Suppression of inflammation and acquired immunity by IL-37). Relevant literature on IL-37 that must be cited and discuss to place the study findings in more proper context include: 

  1. Interleukin 37 reverses the metabolic cost of inflammation, increases oxidative respiration, and improves exercise tolerance (was there any correlation between IL-37 levels and fatigue?)
  2. Treating experimental arthritis with the innate immune inhibitor interleukin-37 reduces joint and systemic inflammation (discuss with regards to arthritis)
  3. Interleukin-37 treatment of mice with metabolic syndrome improves insulin sensitivity and reduces pro-inflammatory cytokine production in adipose tissue (was there any correlation with glycemic indexes or metabolic syndrome, if the data is available?)
  4. Rare genetic variants in interleukin-37 link this anti-inflammatory cytokine to the pathogenesis and treatment of gout (discuss with regards to a common IL-1-mediated pathogenesis shared by AOSD and gout).

Reviewer 3 Report

In the presented manuscript, Nam et al investigated IL-18 and IL-37 serum concentrations in patients with AOSD. While the manuscript is well written in general, its novelty is limited as the serum concentration of both cytokines have been studied in patients with AOSD by several other groups.

Comments:

* A lot of reviews have been cited rather than key original publications.

- Methods:

  • What are the Yamaguchi criteria? Rather than including a reference, the criteria should be listed in the manuscript.
  • The methods need more information. How were ferritin, IL-18 and IL-37 serum concentrations measured? How were other laboratory markers evaluated?

- Results:

  • Given the patients received a variety of anti-inflammatory drugs that would influence the abundance of pro-and anti-inflammatory cytokines, subgroup analyses need to be performed.
  • The statement ‘with borderline significance’ (line 185) should be removed. If p = 0.06, the effect is not significant.
  • Figure 1: The ‘high IL-18’ group seems to be n=9 and not n=10 as indicated in the results. The figure needs a figure legend. Also, it should be stated what the circles depict.
  • The differences between the high IL-37 and high IL-18 groups described in lines 181-188 (which I assume are depicted in the supplementary results which were not available for review) should be moved into the main manuscript as they are the novel findings of this study. However, as commented on earlier, a subgroup analysis needs to show that these findings hold true when accounting for the different medication that patients included in this study received.

- Discussion:

  • Some references are cited incorrectly, e.g. reference [15] which is cited as reference [8] multiple times throughout the manuscript.
  • The difference between the findings in reference 15 and this study need to be discussed more thoroughly. Just to name an example, why did Chi et al find a correlation between IL-37 and IL-18 and this study did not?
  • The authors state that ‘both IL-37 and IL-18 can be useful serologic markers of AOSD activity’, which has been published before and has been cited by the authors themselves. The difference between this and previous studies, in my opinion, is that neither of the 2 cytokines is elevated in all patients with high disease activity and that neither of the markers is a good indicator by themselves but could maybe complement each other. Moreover, according to the observations described in lines 181-188, IL-18 might be a predictive marker for subsequent disease course. In my opinion, this is what the authors should focus/expand on, discuss in more detail (including the discussion of potential reasons why some patients have high IL-37 only and others high IL-18 only) and provide a conclusion for as that would distinguish this manuscript from what is already published.

Round 2

Reviewer 3 Report

I appreciate that Nam et al have addressed all of my comments and concerns and I believe that the revision has significantly improved the manuscript.

Other than a few minor grammatical errors that should be fixed, I have no further concerns.